# Consumers' Value and Risk Perceptions of Circular Fashion: Comparison between Secondhand, Upcycled, and Recycled Clothing

**Inhwa Kim [1,2], Hye Jung Jung [3,\*] and Yuri Lee [1,2,\*]**

1   College of Human Ecology, Seoul National University, Seoul 08826, Korea; eenakim@snu.ac.kr
2   The Research Institute of Human Ecology, Seoul National University, Seoul 08826, Korea
3   DaVinci College of General Education, Chung-Ang University, Seoul 06974, Korea
\*   Correspondence: jayjung@cau.ac.kr (H.J.J.); yulee3@snu.ac.kr (Y.L.)

**Abstract:** A circular economy paradigm has recently emerged to combat environmental pollution and climate change around the world. In the fashion industry, circular fashion has been spotlighted as an environmentally friendly approach to fashion products. The purpose of this study was to investigate consumers' value and risk perceptions, product attitudes, and behavior intentions toward circular fashion consumption. Specifically, this study focuses on three types of circular fashion products from textile waste: reused clothing, upcycled clothing, and recycled clothing. The moderating role of individualism was also explored. Survey data from 850 consumers in their 20s and 30s in Korea were collected to test our hypotheses. The results showed the influence of emotional value was the greatest, while economic risk and performance risk did not affect product attitudes. A moderating effect of individualism was found in the paths between perception dimension and product attitudes and between product attitudes and behavior intention. These findings can help retailers and marketers create more tailored retailing and promotional strategies.

**Keywords:** circular fashion; textile wastes; secondhand clothing; upcycled clothing; recycled clothing; perceived value; perceived risk; individualism; product attitude; behavior intention





## 1. Introduction

The fashion industry is considered one of the most polluting and unsustainable industries with a demoralizing influence on the environment, such as energy consumption, soil, water, and atmospheric systems during the whole process of production, sales, and consumption [1,2]. The fashion industry is responsible for 8 to 10 percent of global carbon emissions (4 to 5 billion tons per year), up to 20 percent of industrial wastewater (79 trillion liters per year) and up to 35 percent of marine microplastic contamination (190,000 tons per year), generating more than 92 million tons of textile waste annually [3]. Even worse, the growth of fast fashion is speeding toward environmental disaster by increasing clothing consumption. In addition, the average number of times a garment is worn before it is discarded has declined [4]. According to the Boston Consulting Group (BCG) and Global Fashion Agenda (GFA), global apparel and footwear consumption from 62 million tons in 2015 will increase by 3.4% per annum to 102 million tons by 2030 [5]. Nonetheless, the textile waste recycling rate is only around 12%. If it remains at this level, 25% of global carbon emissions will occur in the fashion industry by 2050 [6].

As awareness of environmental issues and the potential crisis has grown, national policy changes and eco-friendly consumption have become "essential" rather than "optional." In particular, Millennials and Generation Z (Gen Z), who have recently emerged as major consumers, are showing eco-friendly and ethical consumption behaviors. Several empirical surveys have verified that Millennials and Gen Z worldwide are the most sustainable generations to date [7,8]. Millennials pursue products that are green, ethical, repairable,

long-lasting, and artisanal. Gen Z, who have just begun entering the workforce, also prefer buying sustainable brands and are on par with Millennials. About half of Millennials (50%) and Gen Z (54%) have reported that they are willing to spend an additional 10 percent or even more on sustainable products, which compares to only 34% of Gen X and 23% of Baby Boomers. Since the spending power of Millennials and Gen Z is strong and growing, the concept of sustainability and the circular economy system have been discussed and applied in various sectors and industries. The fashion industry, for example, has been strongly influenced by new approaches to economic development, which has led to a circular fashion paradigm.

The term "circular fashion" is a relatively new concept that combines circular economy and sustainable fashion concepts. A circular economy is not a linear economy based on a "take-make-dispose" model of production but is based on an economy with a closed loop that minimizes environmental pollution by recycling waste and reducing resource consumption [9]. Sustainable fashion is a movement and commitment to fostering a change in fashion products and the fashion system towards greater ecological integrity and social justice [10,11]. Therefore, circular fashion can be defined as clothes, shoes, or accessories that are designed, sourced, produced, and provided with the intention to be used and circulated responsibly and effectively in society for as long as possible in their most valuable form, and thereafter returned safely to the biosphere when they are no longer of human use [12]. In other words, it refers to a regenerative system that takes resource efficiency, non-toxicity, biodegradability, longevity, and recyclability into account from the time the fashion product is designed to when it is disposed so the lifecycle of products does not result in socio-economic loss or environmental damage.

Despite this trend, research on circular fashion has only been conducted at the basic level. Most previous studies have analyzed the characteristics of reused and recycled fashion products from an aesthetic dimension or have analyzed the overall concepts of sustainable fashion such as eco-friendly fashion, eco-fashion, and green fashion from a marketing perspective. Accordingly, it is necessary to broaden our academic and practical understanding by more specifically examining consumers' perceptions of circular fashion. Thus, this study aimed to investigate consumers' perceived values and perceived risks related to circular fashion. Specifically, we divided the types of circular fashion into three sub-dimensions (secondhand clothing, upcycled clothing, recycled clothing) and examined the differences in the impact of consumer perceptions on product attitudes and behavior intention for the three types of circular fashion. This study also aimed to determine the moderating effect of individualism, which is a known general value of Millennials and Gen Z. Marketing research has shown that individual values may be a predictor of consumer behavior since these values are present extensively throughout consumer processes, including the formation of consumer attitudes, information processing, and purchase of goods. Thus, it is expected that individualism could moderate not only the relationship between perceived dimensions and product attitudes but also the relationship between product attitudes and behavior intention. The findings of this research will expand our academic knowledge by providing more in-depth knowledge and a more integrated view of consumers' responses to circular fashion. Furthermore, it will offer insights for retailers by providing information on which values should be emphasized and which risks should be reduced in the sale of circular fashion products.

In the following section, the relevant literature review and the hypotheses development are presented. Subsequently, the research methodology is described, followed by the results. Finally, the results are discussed, and the research contributions, managerial implications, and future research directions are presented.

## 2. Literature Review and Hypotheses Development

### 2.1. Textile Waste Management and Types of Circular Fashion

Textile waste includes raw material, fabric, and clothing or finished products that are deemed no longer useful. Textile waste consists of industry waste generated in the

entire supply chain where fashion and textile products are designed, supplied, produced, distributed, and sold. It also includes consumer waste created by consumer use and the disposal process where fashion and textile products are purchased, damaged, worn out, and discarded. Specifically, textile waste is generated in the following three stages. First, production textile waste can include fiber, yarn, cloth scraps, flock, sweeping, fabric cut-offs, fabric roll ends, and selvage, which are generated primarily during the manufacturing process of clothing and textiles in factories. Second, pre-consumer textile waste is generated during the sale of clothing and textiles at online and offline stores by retailers, such as defective or damaged products and unsold products. Third, post-consumer textile waste can be defined as clothing and textiles that consumers no longer want to use after purchasing, which could be worn-out, damaged, outgrown, or out-of-fashion products.

Given that the amount of textile waste has increased exponentially, researchers, manufacturers, and consumers have paid greater attention to textile waste management and disposal practices. Considerable research has examined textile waste hierarchy. For example, ReThread DC, the District of Columbia's textile reuse and recovery initiative, suggested a textile waste hierarchy to residents and consumers (Figure 1a). First, the most sustainable option is "source reduction" which aims to decrease the accumulation of clothing and other textiles. They proposed that residents shop and swap their own closets to avoid making unnecessary purchases. Second, "reuse" is a way to not damage the original form by reselling unwanted clothing and textiles or by purchasing necessary items from local thrift, resale, and consignment stores. Third, "recovery" extends the product life cycle by repairing clothing and textiles (e.g., patching small holes in pants, sewing on a button that has fallen off). Fourth, "recycle" is a way for unwanted clothing and textiles to not end up in the trash. If items that are dry and clean are donated regardless of their condition, secondary markets will purchase unsellable clothing for use as rags and for industrial fill material. Lastly, "disposal" is the least preferred method of disposal since clothing and textiles end up in a landfill, contributing to the production of lethal gases that cause climate change.

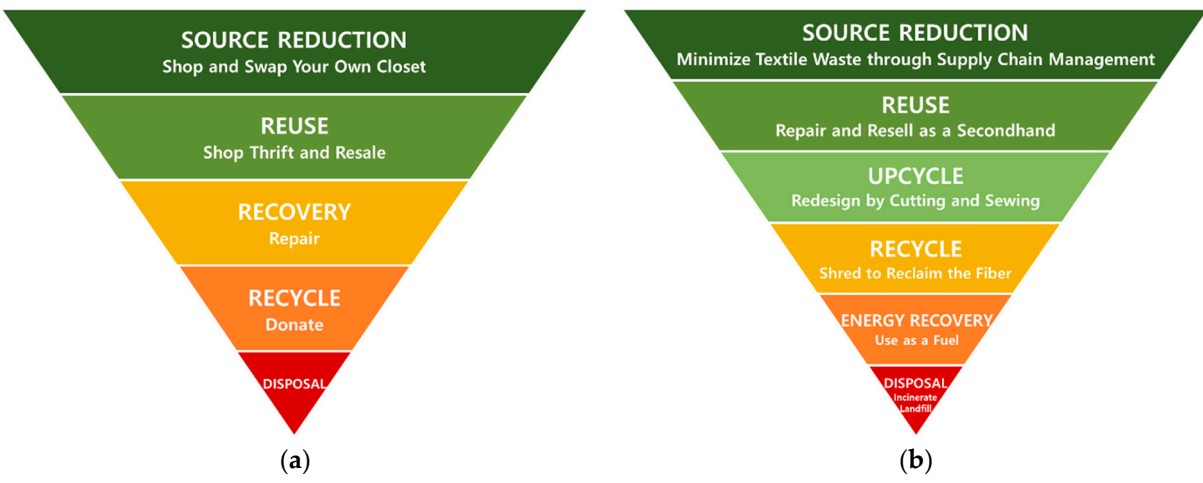

(**a**)　　　　　　　　　　　　　　　　　　　　　　　　(**b**)

**Figure 1.** (**a**) Textile waste hierarchy for consumers; (**b**) textile waste hierarchy for fashion companies.

In this study, we propose a textile waste hierarchy from the perspective of fashion companies (Figure 1b). The top priority is "source reduction," meaning that waste generation should be inhibited through supply chain management by reducing inventory and developing zero-waste designs. "Reuse" means that useable discarded clothing is repaired and re-worn without any transformation. Clothing in good condition collected from a take-back system or unsold stock could be resold as new clothing in the second-hand market. "Upcycle" is a combination of upgrading and recycling, including processes that combine designs and ideas with existing objects to create new uses that are better than the original. For instance, newly designed garments can be produced by cutting and sewing

old clothing, textile scraps, and deadstock. "Recycle" is a method of decomposing textile waste into raw material including reprocessing procedures such as shredding fabric and pulling out fibers. The process often involves leftover fabric and stained or torn clothing and converting it to back to raw material, such as making yarn and fabric, and finally making brand new clothing products. "Energy recovery" is the use of textile waste as a heat or an electricity source, or an alternative fuel instead of fossil fuels. For example, nonrecyclable clothing can be sent to power plants and incinerated to recover energy. Lastly, textile waste can be either landfilled or incinerated as a last resort, which is called "disposal." Reducing the amount of disposed clothing is valuable since the current rate of clothing waste leading to incineration or landfill is the highest. However, disposal does not recover any value from textile waste and has a negative impact on the environment. Based on this textile waste hierarchy, we categorized circular fashion sold in fashion companies into three product types in the order of priority: secondhand clothing, upcycled clothing, and recycled clothing.

### 2.2. Circular Fashion and Consumers' Perceived Value

The basic currency of all human interaction is value, and the only reason consumers engage in consumption behavior is to seek valuable objects [13]. In marketing, value is defined as a cognitive expression of the most basic and fundamental desires and goals that consumers want to obtain [14]. In other words, consumption value is the principal basis for human behavior associated with purchasing and an important constituent of relationship marketing. Therefore, understanding the intrinsic meaning of consumption value can help us understand why consumers choose certain products or brands and why market segmentation is essential [15]. The concept of consumers' perceived value has been defined and adapted in a variety of fields by numerous researchers. Sheth et al.'s [16] theory has been representatively cited in the field of consumer behavior. They incorporated value measures from previous studies by eliminating abstract and irrelevant items and extracting five core consumption values that affect consumer product choices: emotional value, functional value, social value, conditional value, and epistemic value.

In the clothing and textiles field, consumption value research has been specialized and has increasingly been conducted under the concept of clothing consumption value [17–19]. Specifically, researchers have examined the impact of consumer clothing consumption value on the decision-making process or product evaluation of clothing purchases [20]. Jeon et al. [21] classified consumption value of upcycled fashion products into social, emotional, functional, economic, and eco values, and Yu and Lee [22] categorized consumption value into green, functional, emotional, aesthetic, social, and self-expression value. Gallarza et al. [23] suggested that social, emotional, functional, and economic value are the four fundamental dimensions of the perceived value of recycled fashion products. Consistent with these dimensions, Chi [24] divided the perceived value of eco-friendly clothing into social, emotional, functional, and economic value. Choo and Park [25] analyzed the consumption value of secondhand fashion products through in-depth interviews with a group of experts who majored in clothing and textiles and a literature review of prior studies, and identified the categories of rare, emotional, functional, social contribution, economic value.

This study attempted to classify the consumption value of circular fashion using a multidimensional approach based on Sheth et al.'s [16] theory, which was partly modified to take into account the characteristics of circular fashion products without accepting all the value dimensions. Specifically, the functional value in Sheth et al.'s [16] theory refers to the physical and practical consumption value related to the price, quality, and function of a product; however, consumers may be concerned that circular fashion might be less durable and less valuable since the products are made of discarded materials. Therefore, it was ruled out because it was considered a risk factor. The conditional value in Sheth et al.'s [16] theory was also excluded since it is viewed as an alternative based on a specific situation or set of circumstances and thus considered inappropriate in explaining general purchasing

behavior. Instead, we combined the remaining dimensions of environmental value that have been shown to be influential in preceding studies. We employed emotional, social, epistemic, and environmental value as four reflective dimensions of value that consumers can perceive from circular fashion.

Emotional value indicates the perception of value and the emotional state that consumers feel when they shop or wear products [25]. It is the perceived positive emotional changes consumers feel such as pleasure and good feelings when using products. Westbrook and Oliver [26] emphasized the importance of emotional value defining it as a type of hedonic consumption with extreme personal differences, such as pleasure and excitement in experiencing consumption. They argued that feelings of joy and fun and product design in consumption rather than product quality have a greater impact on consumers' choice. From this perspective, consumers relieve stress and feel joy by purchasing circular fashion products that are hedonic goods. Further, consumers feel happy thinking about how they have contributed to environmental protection and that they bought special clothing that is different from non-environmentally friendly clothing. Therefore, it is expected that emotional value will have considerable influence on positive attitude formation toward circular fashion products.

**Hypothesis 1 (H1).** *Emotional value will have a positive effect on consumers' circular fashion product attitude.*

Social value is the perceived utility associated with a particular social group, or the apparent utility that results from the association between the service or product and positively or negatively stereotyped demographic, socioeconomic, and cultural-ethnic groups [16]. In other words, social value of circular fashion is the perception of being socially recognized and receiving a favorable evaluation from others due to the social image of products or the satisfaction of consumers' social needs. If consumers think that circular fashion products are environmentally friendly and scarce and that the products will make a good impression on people around them, the consumer will obtain high social value from the products. In contrast, if consumers worry that they cannot gain social approval since the circular fashion products made of waste may be perceived to be stained or unsanitary, the consumers will obtain low social value from the products. Therefore, this study suggests the following hypothesis.

**Hypothesis 2 (H2).** *Social value will have a positive effect on consumers' circular fashion product attitude.*

Epistemic value is defined as the perceived utility obtained from observers' curiosity about the product's novelty, or their desire for knowledge about the product [16]. The epistemic value associated with circular fashion products is consumers' perception of a product's uniqueness, novelty, or rarity. Numerous studies have verified that secondhand fashion shopping is a treasure hunting activity to discover products that are not readily available in the market [27]. Yoon [28] also explored the purchase motives of reused and recycled fashion products and found that the epistemic value of pursuing unique and rare items had a positive effect on purchase intention of both reused and recycled products. Accordingly, the following hypothesis is formulated.

**Hypothesis 3 (H3).** *Epistemic value will have a positive effect on consumers' circular fashion product attitude.*

Environmental value is a belief about the earth or natural environment and humanity's relationship with it [29]. People with high environmental value lead and develop their lives through the environment. Thus, they believe that they should seek harmony between nature and humans beyond their role in satisfying their own physical or material needs [30]. Environmental value also plays a role in effecting circular fashion consumption because

unlike other fashion products, circular fashion products aim to discourage the use of virgin materials and to encourage the use of recycled materials through closed-loop supply chain management. Numerous studies have concluded that circularity in products [31,32] and a recycled appearance [33,34] have a positive effect on individuals' perceived environmental value. Thus, it is expected that the higher the perceived environmental value, the stronger the product attitude toward circular fashion products.

**Hypothesis 4 (H4).** *Environmental value will have a positive effect on consumers' circular fashion product attitude.*

*2.3. Circular Fashion and Consumers' Perceived Risk*

Perceived risk, first introduced by Bauer [35], is distinct from objective risk. Perceived risk means consumers' perceived anxiety when making a purchase decision about unexpected consequences that may occur after purchasing and using a product [36,37]. Roselius [38] defined the perceived danger in consumer's purchasing behavior as a perceived risk, arguing that when consumers want to buy a product, they face a dilemma between the loss incurred after the purchase and the desire to buy it. Research on perceived risk has a long history. Most research on the influence of risk perception has used a multidimensional approach rather than a unidimensional approach to comprehensively explain the risk factors in the customer's purchasing decision process [39–41]. Although there are some differences among researchers, a lower dimension of perceived risk is generally divided into social, psychological, financial, and functional risk [42,43].

Many studies have been conducted on the perceived risks in the field of clothing and textiles. Koyama et al. [44] surveyed participants on the perceived risks associated with clothing products and identified five anxiety factors: anxiety about quality/performance, anxiety about deviating from clothing norms, anxiety about looking good in style, anxiety about being current, and anxiety about being too showy. Compared to earlier research on perceived risks, these five anxiety factors are distinct empirical classifications of perceived fashion risk [45]. In terms of the perceived risks of circular fashion, Park and Choo [46] conducted a qualitative study to derive the perceived risk dimension of upcycled fashion products, and identified five dimensions: aesthetic, sanitary, social, performance, and economic risk. Kim and Kim [47] also suggested that diversity risk and performance risk are two fundamental dimensions of perceived risk of upcycled products, while Yoon [28] proposed performance risk, economic risk, and social psychological risk as perceived risks of reused and recycled clothing.

In the current study, we classified the risk dimensions of circular fashion based on Park and Choo [46] since they took a similar stance to ours Among their five risk perception dimensions, social risk (i.e., anxiety about being judged by others about the design and appearance of recycled clothing) was excluded because it somewhat overlapped with the explanation of aesthetic risk and because several studies have shown that it is not associated with word of mouth (WOM) intention and purchase intention. Thus, we divided consumers' perceived risks of circular fashion into financial, functional, aesthetic, and sanitary risk in this study, and supported each risk dimension with reference to previous studies.

Financial risk refers to consumers' perceived economic risk of investment loss or additional costs for repair or replacement of the purchased product when there is a problem [48,49]. Consumers considering purchase of circular fashion might also perceive financial risk in that they think circular fashion products would be relatively expensive compared to general fashion products since they cannot be mass-produced, or circular fashion products would be sold at high prices in light of various conditions. Park and Choo [46] found that the majority of purchasers had a negative perception that the price of upcycled fashion products was too expensive. Further, Machado et al. [27] found that unlike the purpose of secondhand stores, recently used fashion products are sold at high prices, which is seen as an unethical behavior conducted by thrift shop owners. Thus,

consumers perceive a financial risk of a circular fashion product and may have a negative attitude toward the products.

**Hypothesis 5 (H5).** *Financial risk will have a negative effect on circular fashion product attitudes.*

Functional risk, also known as performance risk, is the uncertainty associated with the consequence of a product that does not function as expected [48,50]. In other words, functional risk refers to the possibility that the purchased product will not work well, the service will not be performed properly, or the desired function will fail [42,51,52]. Circular fashion products are clearly distinct from general products in that they go through the process of being newly designed and manufactured and sold by recycling discarded materials. The use of wasted materials or products and an unusual production process may make people perceive the functional uncertainty of circular fashion products, and thus postpone purchase behavior. Park and Choo [46] contended that due to the specificity of upcycling, consumers may be worried that upcycled products will not fulfill the utilitarian needs of consumers such as durability, effectiveness, ease of use, and sturdiness. Based on this reasoning, the following hypothesis is formulated.

**Hypothesis 6 (H6).** *Functional risk will have a negative effect on consumers' circular fashion product attitude.*

Aesthetic risk is the perception that the purchased product will not be in line with the consumer's self-image. Specifically, perceived aesthetic risk related to clothing purchases include harmony with one's image as well as harmony with other clothing the consumer owns, implying that clothing plays an important role in aesthetically satisfying consumers' need for a congruent self-image [53]. Likewise, aesthetic risk of circular fashion clothing refers to apprehension about whether circular fashion clothing is well coordinated with the other clothing the consumer owns such as a lack of diversity, whether the product matches well with the consumer's image, or whether there is a poor size fit. That is, consumers might be concerned that recycled fashion products are not diverse enough in color, size, and style. They might also be concerned that the secondhand clothing that others have worn and discarded is out of fashion. Kim [54] found that consumers' perceived aesthetic risk could mean delaying or abandoning the purchase. In other words, consumers might perceive an aesthetic risk of circular fashion products and thus they are less likely to prefer such products.

**Hypothesis 7 (H7).** *Aesthetic risk will have a negative effect on consumers' circular fashion product attitude.*

Lastly, sanitary risk is defined as the perception of anxiety that might disrupt one's life or harm one's health. In this study, we designated sanitary risk as the perceived anxiety that circular fashion products are not clean and are likely to have stains or dirt which could defile the body when wearing the products. Put simply, sanitary risk of circular fashion clothing reflects consumers' perception of the possibility that circular fashion clothing made from waste or discarded materials might not be hygienic or brand new. According to a study on the reutilization of used school uniforms of middle and high school students, 18% of the subjects responded that they had not bought used school uniforms because of potential hygiene issues [55]. Qualitative research to identify perceived risk dimensions of upcycled products also showed that consumers who had purchased upcycled products were sensitive to other people's negative views or opinions about the sanitation of the products [46]. In particular, unlike other types of products, clothing directly touches the body, so sanitary risk is expected to wield even greater influence over consumers' product attitude.

**Hypothesis 8 (H8).** *Sanitary risk will have a negative effect on consumers' circular fashion product attitude.*

### 2.4. Product Attitude and Behavior Intention

Engel and Blackwell [56] defined attitude as the basic direction of personal likes and dislikes toward people, objects, and phenomena, suggesting that it is the basis for consumer behavior. Based on this concept of attitude, product attitude in this study refers to consumers' overall positive or negative assessment of circular fashion clothing. As social beings, consumers tend to exchange information about their purchase experiences with other consumers and their purchasing decisions reflect other people's experiences. This concept is defined as WOM. In this study, WOM intention refers to the intention to share positive feelings and information about circular fashion clothing with others and recommend it to others. In other words, this study focuses on positive WOM. Purchase intention refers to the expected or planned future behavior of consumers, which means the likelihood that consumers are willing to purchase a certain product or service in the future [56,57]. We designated purchase intention as consumers' willingness to buy circular fashion products in the near future.

Consumers have regular behavioral patterns in which they recognize products through subjective emotions and surrounding socio-environmental factors. They form beliefs, explore and evaluate relevant information, and then form an attitude resulting in specific actions. Numerous studies have examined sustainable consumption behavior and confirmed that a consumer's product attitude is a key factor that triggers actual consumer behavior [58–60]. For example, Kang et al. [59] verified that consumers' product attitude has considerable influence on behavior intention in the context of environmentally sustainable fashion consumption. Park and Oh [60] also found that consumers' attitude about eco-friendly fashion products has a positive effect on both WOM intention and purchase intention. Likewise, it is expected that consumers' attitude toward sustainable fashion clothing will positively affect WOM intention and purchase intention. WOM refers to recommending goods or services to acquaintances and does not require financial costs. WOM intention only refers to the possibility of delivering information associated with a certain product or service with which the consumer is satisfied. The consumer has no entangled interest in the company and does not benefit from the company. Although a purchase is deemed to be a more powerful consumer behavior in that the consumer incurs a financial cost to buy the product or enjoy the service, it is generally assumed that the higher the WOM intention, the higher the purchase intention.

**Hypothesis 9 (H9).** *Consumers' positive product attitude toward circular fashion will have a positive effect on WOM intention.*

**Hypothesis 10 (H10).** *Consumers' positive product attitude toward circular fashion will have a positive effect on purchase intention.*

**Hypothesis 11 (H11).** *WOM intention will have a positive effect on consumers' purchase intention.*

### 2.5. Individualism as a Moderating Variable

Millennials and Gen Z (MZ generation, hereafter) has become an important target of the fashion industry's marketing strategy. Thus, we include individualism in this study given that the peculiarity of MZ generation affects circular fashion consumption. Individualism prioritizes the interests of individuals rather than the goals of the group, and thus emphasizes the right to decide one's own actions [61]. Hofstede [62] refers to individualism as a state of emotional independence from groups, organizations, and other aggregates. Hui [63] also noted that individualists are people who define themselves as being independent of groups. In other words, individualism is an attitude in which individuals take precedence over groups, and people think and make decisions alone rather

than relying on their surroundings. In most cases, it also emphasizes individual identity, interests, and independence.

People with strong individualism are inclined to value their own interests, purposes, experiences, and values rather than social situations [64]. From this point of view, it seems that individualists are less interested in circular fashion because they are less sensitive to the social environment. However, Marcus and Kitayama [65] indicated that just because those with high individualism prioritize self as a frame of reference, it does not follow that they always make independent decisions regardless of their surroundings. They argued that as social beings, it is natural to seek relationships with group members and all social beings are interested in the social environment and issues. Social identity theory, which contends that in-group homogeneity is crucial, supports this stance that individualists' characteristic of self-reliance and uniqueness could affect the self-identity of individualists, prioritizing their affiliated group over social identity. People with high individualism may also realize that they can personally benefit from practicing sustainable behavior such as saving the earth and protecting the planet. Thus, they may be willing to purchase circular fashion products for personal gain. Barbarossa and De Pelsmacker [66] also supported this linkage by emphasizing ego-centric motivations (i.e., green self-identity and moral obligation) based on psychological egoism theory. It is also expected that once those with a strong disposition toward individualism form a positive attitude toward circular fashion, they may further strengthen their WOM intention and purchase intention, since they value their own judgement in behavior decisions. Hence, we anticipate that individualism has a positive effect on circular fashion consumption.

**Hypothesis 12 (H12).** *Individualism will have a moderating effect on the relationship between consumers' perception and attitude toward circular fashion.*

**Hypothesis 13 (H13).** *Individualism will have a moderating effect on the relationship between consumers' attitude toward circular fashion and behavior intention.*

The conceptual model of this study is presented in the Figure 2.

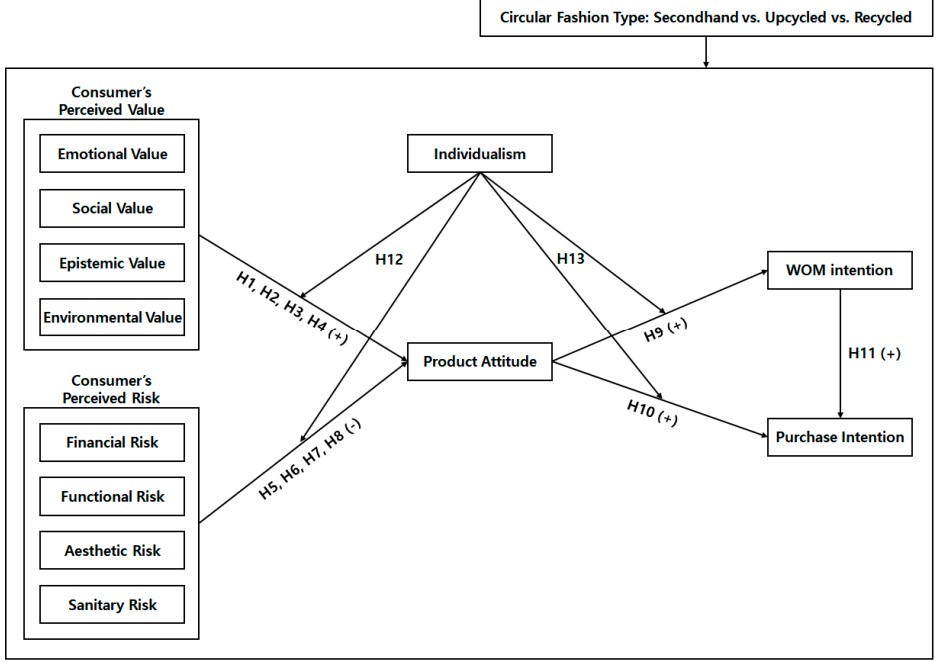

**Figure 2.** Conceptual model.

## 3. Method

### 3.1. Procedure and Stimuli Development

This study designed a questionnaire using stimuli. The questionnaire consisted of a between-subjects experimental design with three circular fashion types: (1) secondhand clothing (reusing textile waste in its original form or after washing and repairing), (2) upcycled clothing (upcycling textile waste through redesign), and (3) recycled clothing (recycling textile waste by shredding to reclaim the fibers). At the beginning of the survey, we provided a brief explanation of textile waste and a description of all three types of circular fashion production processes. Participants were then asked to choose their favorite type of circular fashion clothing. Next, respondents were randomly assigned to one of the three types of circular fashion clothing. To clearly differentiate the type they would respond to against the other two types, a description of the type and two examples of fashion brands were provided. After a manipulation check, the full-scale questionnaire was administered.

For the stimuli development, we provided visuals of all three types of production processes using Adobe Illustrator (see Appendix A). By expressing each type as drawings rather than real pictures, we could control the recognition and popularity of specific brands or products and exclude influence from sources other than the research variables. We first presented the stimuli to 20 people majoring in clothing and textiles to confirm our choices. They found that the drawings and photographs along with the brief description made it easier to understand the content and that there was no difficulty in distinguishing the differences between the three types. Their feedback also confirmed that the stimuli in this study was appropriate.

### 3.2. Measurements

The measurement items for this study were adopted from prior studies (Table 1). The questionnaire consisted of scales that measure perceived value (emotional value, social value, epistemic value, environmental value), perceived risk (economic risk, performance risk, aesthetic risk, sanitary risk), product attitude, behavior intention (WOM intention, purchase intention), and individualism as a personal characteristic. All items on perceived value were adapted from Sheth et al. [16] and Yu and Lee [22]. Consumers' product attitude towards circular fashion clothing was measured with the same scale from Yu and Lee [22]. The perceived risk scale and WOM intention scale were borrowed from Park and Choo [46]. To measure purchase intention, the scale was taken from Kim and Kim [47]. The items of individualism were based on Yoon and Kim [67] and Han and Na [68]. A seven-point Likert scale was used to measure all multi-item scales.

**Table 1.** Results of exploratory factor analysis and reliability check.

| Factor | Items | Factor Loading | Eigen Value | % of Variance | Cronbach's α |
|---|---|---|---|---|---|
| Emotional value | 1. I feel happy when I wear this clothing. | 0.866 | 3.690 | 12.299 (12.299) | 0.914 |
| | 2. Purchasing this clothing makes me feel good. | 0.854 | | | |
| | 3. The stress is relieved by purchasing this clothing. | 0.831 | | | |
| | 4. This clothing provides joy and pleasure. | 0.805 | | | |
| Social value | 5. Purchasing this clothing can give its owner social approval. | 0.726 | 2.005 | 6.682 (18.981) | 0.841 |
| | 6. This clothing would make a good impression on other people. | 0.718 | | | |
| | 7. This clothing would improve the way I am perceived by my friends. | 0.711 | | | |

**Table 1.** *Cont.*

| Factor | Items | Factor Loading | Eigen Value | % of Variance | Cronbach's α |
|---|---|---|---|---|---|
| Epistemic value | 8. This clothing offers uniqueness. | 0.802 | 2.232 | 7.441 (26.422) | 0.889 |
| | 9. This clothing has points of difference from general clothing. | 0.793 | | | |
| | 10. This clothing has many new features. | 0.690 | | | |
| Environmental value | 11. This clothing helps save resources. | 0.919 | 3.519 | 11.731 (38.153) | 0.934 |
| | 12. This clothing has a positive impact on the environment in that it extends the life of discarded materials. | 0.905 | | | |
| | 13. This clothing is environmentally friendly. | 0.875 | | | |
| | 14. This clothing has more environmental benefits than other clothing. | 0.869 | | | |
| Financial risk | 15. This clothing is likely to be expensive in light of various conditions. | 0.910 | 3.285 | 10.949 (49.102) | 0.903 |
| | 16. This clothing is likely to be expensive because it is not mass-produced. | 0.900 | | | |
| | 17. This clothing is likely to be relatively expensive compared to general clothing. | 0.884 | | | |
| | 18. There would be a price bubble in this clothing. | 0.740 | | | |
| Functional risk | 19. This clothing would not be durable. | 0.867 | 3.312 | 11.041 (60.143) | 0.909 |
| | 20. I won't be able to wear this clothing for a long time. | 0.854 | | | |
| | 21. This clothing is likely to wear out faster than general clothing. | 0.841 | | | |
| | 22. This clothing is likely to damage its style or color if washed. | 0.760 | | | |
| Aesthetic risk | 23. This clothing would not fit well because it does not vary in size. | 0.758 | 2.494 | 8.314 (68.457) | 0.800 |
| | 24. This clothing would not vary in design. | 0.727 | | | |
| | 25. This clothing would not reflect the latest trends in design or style. | 0.711 | | | |
| | 26. This clothing would not be easy to coordinate with other clothing. | 0.703 | | | |
| Sanitary risk | 27. This clothing is unlikely to be clean. | 0.885 | 3.327 | 11.089 (79.546) | 0.932 |
| | 28. This clothing is likely to have stains or dirt. | 0.877 | | | |
| | 29. This clothing is unlikely to be hygienic. | 0.842 | | | |
| | 30. This clothing would not be seen as a new product. | 0.692 | | | |
| Product attitude | 31. I like this clothing. | 0.960 | 3.502 | 87.545 | 0.952 |
| | 32. I have a positive emotion regarding this clothing. | 0.954 | | | |
| | 33. I am interested in this clothing. | 0.931 | | | |
| | 34. I think positively about this clothing. | 0.896 | | | |
| WOM intention | 35. I want to introduce this clothing to people around me. | 0.873 | 2.997 | 49.947 (49.947) | 0.951 |
| | 36. I am willing to recommend this clothing to people around me. | 0.857 | | | |
| | 37. I want to tell others about the experience and feeling of wearing this clothing. | 0.748 | | | |
| | 38. If someone asks me for advice on this clothing, I would highly recommend it. | 0.736 | | | |

**Table 1.** *Cont.*

| Factor | Items | Factor Loading | Eigen Value | % of Variance | Cronbach's α |
|---|---|---|---|---|---|
| Purchase intention | 39. I would like to buy this clothing. | 0.871 | 2.396 | 39.938 (89.885) | 0.942 |
| | 40. I am willing to buy this clothing when I shop my clothing in the near future. | 0.850 | | | |
| Individualism | 41. I live my life in my own way. | 0.848 | 1.820 | 60.664 | 0.658 |
| | 42. It is important for me to pursue my own personality. | 0.836 | | | |
| | 43. I make decisions and act in consideration of my own interests rather than the interests of the group. | 0.635 | | | |

*3.3. Sample*

Korean MZ generation participants, born between 1985 and 2001, were recruited through a professional research firm and the online survey was conducted in April 2020. Many studies have suggested that the MZ generation will lead consumption over the next 20 years, and that they are interested in sustainable products and value-based consumption. Considering this trend, this study focused on the MZ generation. After excluding individuals who provided insincere answers, a total sample of 850 participants was available for data analysis (secondhand clothing, $n$ = 289; upcycled clothing, $n$ = 282; recycled clothing, $n$ = 279).

The sample consisted of 49.3% ($n$ = 419) males, 50.7% ($n$ = 431) females, and 50.2% ($n$ = 427) Millennial generation, 49.8% ($n$ = 423) Z generation. Most participants lived in the metropolitan areas of Seoul (30.1%) and Gyeonggi (27.6%). In terms of education level, 50.5% held a bachelor's degree, 35.5% were studying in a university, 7.3% had a master's or higher, 6.6% completed high school or less, and 0.1% identified as neither. For the average monthly income of households, 23.1% earned between ₩2,000,000 and ₩3,000,000, 16.0% earned between ₩3,000,000 and ₩4,000,000, and 14.0% earned between ₩6,000,000 and ₩8,000,000. Concerning average monthly fashion expenditure, between ₩100,000 and ₩300,000 was the most common (41.6%), followed by between ₩50,000 and ₩100,000 (30.0%), less than ₩50,000 (16.0%), and other (12.3%).

**4. Results**

*4.1. Measurement Reliability*

We identified the factorial structures of all variables using exploratory factor analysis (EFA; principal factor analysis with varimax rotation). Factor analysis of consumer perception of circular fashion generated eight factors, and factor analysis of consumer behavior intention generated two factors. All factor loadings were confirmed to be from 0.635 to 0.960, exceeding the corresponding cutoff criteria. The eigenvalues for the factors were all above 1.0, indicating acceptable construct validity of each variable [69]. In addition, the Cronbach's α values for the items in each factor ranged from 0.658 to 0.952, indicating that the items in each factor were internally consistent. Thus, convergent and discriminant validity were adequate for our measurement model [69] (see Table 1).

*4.2. Manipulation Check and Homogeneity Test*

Prior to administering the full questionnaire, respondents were asked to answer two manipulation check questions to ensure that they could rate only their assigned circular fashion type and not overall circular fashion. Participants who answered one or both questions wrong were excluded, but most participants in this study were aware of the assigned type they should respond to and they could clearly distinguish it from the other two types.

A one-way ANOVA was conducted to confirm that there was no difference in demographic characteristics of the participants between the three comparative groups. There was no difference found between groups in all demographic variables including gender, generation, education level, occupation, and monthly average household income. We also conducted a one-way ANOVA to check homogeneity of individualism between the groups since we needed to control the external effects that would result from differences in participants' sense of value. The results showed no significant differences between groups in homogeneity of individualism.

### 4.3. Hypothesis Testing

A regression analysis for the three types of circular fashion, respectively, and the group as a whole were conducted to examine H1 through H11 using SPSS 25.0. PROCESS Macro Model 1 was used to determine the moderating effect of individualism for H12 and H13. Table 2 summarizes the results of the hypothesized relationships.

**Table 2.** Results of hypothesis testing.

| H | Path | The Entire Circular Fashion Clothing (*n* = 850) | Group A. Secondhand Clothing (*n* = 289) | Group B. Upcycled Clothing (*n* = 282) | Group C. Recycled Clothing (*n* = 279) |
|---|---|---|---|---|---|
| H1 | EMV → PA | Accepted (+) | Accepted (+) | Accepted (+) | Accepted (+) |
| H2 | SOV → PA | Accepted (+) | Accepted (+) | Accepted (+) | Accepted (+) |
| H3 | EPV → PA | Accepted (+) | Accepted (+) | Accepted (+) | X |
| H4 | ENV → PA | Accepted (+) | Accepted (+) | x | Accepted (+) |
| H5 | FIR → PA | x | x | x | x |
| H6 | FUR → PA | x | x | x | x |
| H7 | AER → PA | Accepted (−) | Accepted (−) | x | Accepted (−) |
| H12 | AER → PA (AER X ID) | x | ID strengthened negative effect | x | x |
| H8 | SAR → PA | Accepted (−) | Accepted (−) | Accepted (−) | Accepted (−) |
| H12 | SAR → PA (SAR X ID) | ID weakened negative effect | ID weakened negative effect | x | x |
| H9 | PA → WOM | Accepted (+) | Accepted (+) | Accepted (+) | Accepted (+) |
| H13 | PA → WOM (PA X ID) | ID strengthened positive effect | X | ID strengthened positive effect | x |
| H10 | PA → PI | Accepted (+) | Accepted (+) | Accepted (+) | Accepted (+) |
| H13 | PA → PI (PA X ID) | ID strengthened positive effect | X | ID strengthened positive effect | x |
| H11 | WOM → PI | Accepted (+) | Accepted (+) | Accepted (+) | Accepted (+) |

Notes: EMV: Emotional Value; SOV: Social Value; EPV: Epistemic Value; ENV: Environmental Value; FIR: Financial Risk; FUR: Functional Risk; AER: Aesthetic Risk; SAR: Sanitary Risk; PA: Product Attitude; WOM: Word of Mouth Intention; PI: Purchase Intention; ID: individualism.

#### 4.3.1. The Effect of Value and Risk Perceptions of Circular Fashion on Product Attitude

We conducted multiple regression analysis to investigate the effect of the perceived dimensions of circular fashion on product attitude. The results of the entire group are as follows. The regression model was significant (F-value = 189.999, $p < 0.001$), and explained 64% of the variance in product attitude. It showed that emotional value ($\beta = 0.390$, $p < 0.001$), social value ($\beta = 0.196$, $p < 0.001$), epistemic value ($\beta = 0.109$, $p < 0.001$), and environmental value ($\beta = 0.125$, $p < 0.001$) had a significant positive influence on product attitude, whereas aesthetic risk ($\beta = -0.113$, $p < 0.001$) and sanitary risk ($\beta = -0.224$, $p < 0.001$) had a significant negative influence on product attitude. No significant effect was found for

financial risk ($\beta = 0.005$, $p > 0.05$) and functional risk ($\beta = 0.017$, $p > 0.05$). Thus, H1, H2, H3, H4, H7, and H8 were supported and H5 and H6 were not supported.

In terms of the differences between groups, for secondhand clothing, the results are as follows. The resulting indices indicated the significance of the regression model (F-value = 55.331, $p < 0.001$) with 61% explanatory power. Emotional value ($\beta = 0.417$, $p < 0.001$), social value ($\beta = 0.182$, $p < 0.001$), epistemic value ($\beta = 0.171$, $p < 0.001$), and environmental value ($\beta = 0.116$, $p < 0.01$) had a significant positive influence on product attitude, thus supporting H1, H2, H3, and H4. Aesthetic risk ($\beta = -0.122$, $p < 0.01$) and sanitary risk ($\beta = -0.169$, $p < 0.001$) had a negative influence on product attitude but financial risk ($\beta = -0.004$, $p > 0.05$) and functional risk ($\beta = 0.034$, $p > 0.05$) had no significant effect. Thus, H5 and H6 were rejected and H7 and H8 were supported.

The results of upcycled clothing are as follows. The regression model was significant (F-value = 71.637, $p < 0.001$), and explained 68% of the variance in product attitude. For upcycled clothing, emotional value ($\beta = 0.379$, $p < 0.001$), social value ($\beta = 0.253$, $p < 0.001$), and epistemic value ($\beta = 0.183$, $p < 0.01$) positively affected product attitude, while sanitary risk ($\beta = -0.223$, $p < 0.001$) negatively affected product attitude. Therefore, H1, H2, H3, and H8 were supported. Environmental value ($\beta = 0.030$, $p > 0.05$), financial risk ($\beta = -0.011$, $p > 0.05$), functional risk ($\beta = -0.012$, $p > 0.05$), and aesthetic risk ($\beta = -0.073$, $p > 0.05$) had no significant influence on product attitude. Thus, H4, H5, H6, and H7 were rejected.

As for the group of recycled clothing, the regression model was significant (F-value = 50.375, $p < 0.001$), and explained 60% of the variance in product attitude. Emotional value ($\beta = 0.373$, $p < 0.001$), social value ($\beta = 0.117$, $p < 0.05$), and environmental value ($\beta = 0.201$, $p < 0.001$) positively affected product attitude except for epistemic value ($\beta = 0.112$, $p > 0.05$). However, aesthetic risk ($\beta = -0.115$, $p < 0.05$) and sanitary risk ($\beta = -0.278$, $p < 0.001$) negatively affected product attitude and the effects of financial risk ($\beta = 0.064$, $p > 0.05$) and functional risk ($\beta = 0.096$, $p > 0.05$) were not significant. Therefore, H1, H2, H4, H7, and H8 were supported and H3, H5, and H6 were not supported.

### 4.3.2. The Effect of Product Attitude towards Circular Fashion on Behavior Intention

We conducted a simple linear regression analysis for each type to verify the causal relationships between product attitude and behavior intention. First, the analysis on the effect of product attitude towards circular fashion on WOM intention showed that the F-value representing the fit of the regression model was 1377.883 for all circular fashion, 460.640 for the secondhand clothing group, 545.748 for the upcycled clothing group, and 276.475 for the recycled clothing group. All were significant at the level of $p < 0.001$. The strong positive influence of the product attitude on WOM intention was identified in all four groups ($\beta \geq 0.707$, $p < 0.001$), thus supporting H9.

The analysis of the effect of product attitude towards circular fashion on purchase intention showed that the F-value indicating the goodness of model fit was 1410.645 for all circular fashion, 529.810 for the secondhand clothing group, 519.434 for the upcycled clothing group, and 316.434 for the recycled clothing group. They were all significant at the level of $p < 0.001$. Product attitude had a positive and significant effect on purchase intention in all four groups ($\beta \geq 0.730$, $p < 0.001$), as we hypothesized, thus supporting H10.

The analysis of the effect of WOM intention on purchase intention indicated that the F-value representing the fit of the regression model was 2057.337 for all circular fashion, 644.473 for the secondhand clothing group, 716.232 for the upcycled clothing group, and 598.486 for the recycled clothing group. They were all significant at the level of $p < 0.001$. WOM intention had a strong and positive influence on purchase intention in all four groups ($\beta \geq 0.827$, $p < 0.001$), thus supporting H11.

### 4.3.3. Moderating Effects of Individualism

The bootstrap method was applied to identify the moderating effects of individualism in the relationship between perceived dimensions and product attitude, and between

product attitude and behavior intention. Specifically, PROCESS Macro Model 1 was conducted by designating 95% of the confidence intervals and 5000 bootstrap samples. If 0 was included between the lower limit confidence interval (LLCI) and the upper limit confidence interval (ULCI), the null hypothesis was supported. However, if not, the alternative hypothesis was supported, indicating that there was a moderating effect. The bootstrap results are as follows.

Regarding the entire circular fashion group, a moderating effect of individualism was found in the relationship between sanitary risk and product attitude (B = 0.088, 95% CI = 0.0355~0.1409). That is, the higher the level of individualism, the weaker the negative effect of sanitary risk on product attitude. In addition, we also found moderating effects of individualism in the relationship between product attitude and behavior intention. The result showed that the higher the level of individualism, the stronger the positive impact of product attitude on WOM intention (B = 0.064, 95% CI = 0.0161~0.1113). It also showed that the higher the level of individualism, the stronger the positive impact of product attitude on purchase intention (B = 0.058, 95% CI = 0.0087~0.1063). Thus, H12 was partially supported and H13 was fully supported.

As for the secondhand clothing group, the moderating effect was confirmed in the path from aesthetic risk to product attitude and from sanitary risk to product attitude. More specifically, the result indicated that the higher the level of individualism, the stronger the negative impact of aesthetic risk on product attitude (B = −0.143, 95% CI = −0.2792~−0.0067). However, the result revealed that the higher the level of individualism, the weaker the negative impact of sanitary risk on product attitude (B = 0.186, 95% CI = 0.0790~0.2922). Thus, H12 was partially supported and H13 was rejected.

For the upcycled clothing group, there were moderating effects in the relationship between product attitude and behavior intention. The result showed that the higher the level of individualism, the stronger the positive impact of product attitude on WOM intention (B = 0.078, 95% CI = 0.0083~0.1473). It also indicated that the higher the level of individualism, the stronger the positive impact of product attitude on purchase intention (B = 0.078, 95% CI = 0.0034~0.1523). Therefore, H12 was rejected and H13 was accepted. No moderating effect of individualism was found in the recycled clothing group, rejecting both H12 and H13.

### 4.4. Between-Group Differences in Variable Means: Secondhand Clothing vs. Upcycled Clothing vs. Recycled Clothing

A one-way ANOVA was conducted to confirm whether the mean differences of the sub-variables was significant, depending on the type of circular fashion. As a result, significant mean differences were found among the three groups in all dependent variables. Next, a post hoc analysis was conducted to determine the differences among the three groups. The results are shown in Table 3. Overall, MZ generation perceived a lower value and higher risk for secondhand clothing than upcycled clothing and recycled clothing. However, secondhand clothing showed the lowest mean value only for financial risk, which measured anxiety over expensive prices. In contrast, recycled clothing showed a high value and low risk compared to the other two types, and the mean of both product attitude and behavior intention were significantly higher.

**Table 3.** Results of ANOVA and post hoc analysis.

| Variable | Group | Mean | Standard Deviation | F | Result |
|---|---|---|---|---|---|
| Emotional Value | Secondhand Clothing (a) | 4.0234 | 1.14189 | 10.207 *** | c,b > a (Scheffe) |
| | Upcycled Clothing (b) | 4.3768 | 1.07921 | | |
| | Recycled Clothing (c) | 4.3916 | 1.08709 | | |
| Social Value | Secondhand Clothing (a) | 4.0288 | 1.12951 | 25.711 *** | c > b > a (Scheffe) |
| | Upcycled Clothing (b) | 4.3948 | 1.07791 | | |
| | Recycled Clothing (c) | 4.6977 | 1.13261 | | |
| Epistemic Value | Secondhand Clothing (a) | 4.2307 | 1.24282 | 96.586 *** | b > c > a (Dunnett T3) |
| | Upcycled Clothing (b) | 5.5520 | 1.01970 | | |
| | Recycled Clothing (c) | 5.0370 | 1.07019 | | |
| Environmental Value | Secondhand Clothing (a) | 5.8253 | 1.03044 | 7.612 ** | c,b > b,a (Scheffe) |
| | Upcycled Clothing (b) | 5.9512 | 0.90401 | | |
| | Recycled Clothing (c) | 6.1362 | 0.92078 | | |
| Financial Risk | Secondhand Clothing (a) | 2.8382 | 1.20977 | 71.290 *** | b > c > a (Dunnett T3) |
| | Upcycled Clothing (b) | 4.1144 | 1.36279 | | |
| | Recycled Clothing (c) | 3.5493 | 1.22568 | | |
| Functional Risk | Secondhand Clothing (a) | 4.3054 | 1.12518 | 37.227 *** | b,a > c (Dunnett T3) |
| | Upcycled Clothing (b) | 4.4105 | 1.25354 | | |
| | Recycled Clothing (c) | 3.5672 | 1.28589 | | |
| Aesthetic Risk | Secondhand Clothing (a) | 3.9905 | 1.03441 | 69.388 *** | a > b > c (Dunnett T3) |
| | Upcycled Clothing (b) | 3.6055 | 1.16831 | | |
| | Recycled Clothing (c) | 2.8987 | 1.17368 | | |
| Sanitary Risk | Secondhand Clothing (a) | 4.2898 | 1.25888 | 70.140 *** | a > b > c (Scheffe) |
| | Upcycled Clothing (b) | 3.6764 | 1.36903 | | |
| | Recycled Clothing (c) | 3.0000 | 1.26271 | | |
| Product Attitude | Secondhand Clothing (a) | 4.3746 | 1.23377 | 36.804 *** | c > b > a (Scheffe) |
| | Upcycled Clothing (b) | 4.7766 | 1.21662 | | |
| | Recycled Clothing (c) | 5.2276 | 1.09761 | | |
| WOM Intention | Secondhand Clothing (a) | 3.9196 | 1.34976 | 25.636 *** | c > b > a (Scheffe) |
| | Upcycled Clothing (b) | 4.2012 | 1.32877 | | |
| | Recycled Clothing (c) | 4.6900 | 1.19979 | | |
| Purchase Intention | Secondhand Clothing (a) | 4.0277 | 1.42617 | 18.771 *** | c > b,a (Dunnett T3) |
| | Upcycled Clothing (b) | 4.1152 | 1.40348 | | |
| | Recycled Clothing (c) | 4.6398 | 1.20558 | | |

** $p < 0.01$, *** $p < 0.001$

## 5. Discussion

The aim of our study was to investigate consumers' value and risk perceptions of circular fashion. We divided circular fashion into three types based on a textile waste hierarchy and examined whether the dimensions of the perceived value and perceived risk influenced product attitude and behavior intention. Further, we tested the moderating effects of individualism in all paths. The results of the study revealed several key findings.

First, the results of multiple regression analysis on circular fashion product attitude showed that perceived value plays a more important role in product attitude formation than perceived risk. This finding suggests that perceived value dimensions in circular fashion have considerable influence on product attitude, which can create a positive product attitude even if consumers perceive risks. In particular, the findings confirm that the influence of emotional value (i.e., feeling joy and pleasure) from circular fashion clothing is the greatest, and financial risk and functional risk do not significantly affect consumers' product attitude. Among the risk factors, sanitary risk (i.e., a concern about the possibility of products not being hygienic) had the greatest negative impact on consumers'

product attitude. In other words, high anxiety about price and durability does not lead to a more negative assessment of circular fashion clothing. However, perceiving that circular fashion clothing is unlikely to be clean contributes greatly to the formation of a negative product attitude due to the nature of clothing products that come in close contact with the body. Specifically, the results of comparing the effects of perceived dimensions on product attitude by type of circular fashion clothing showed that for all types, emotional value and social value had a positive influence on product attitude, and sanitary risk had a negative influence on product attitude. Additionally, for secondhand clothing, epistemic value and environmental value had a positive effect on product attitude, and a concern about possible discrepancies with the latest trends had a negative impact on product attitude. For upcycled clothing, the results indicated that perceiving environmental benefits had no positive impact on product attitude, and aesthetic risk had no negative impact on product attitude. Regarding recycled clothing, recognizing the uniqueness or rarity of the product did not have a positive effect on product attitude, but environmental benefits were an important factor in the formulation of product attitude.

Next, in terms of the effect of product attitude on behavior intentions, the same results were found in all groups. As expected, product attitude had a positive effect on WOM intention and purchase intention, and WOM intention had a positive effect on purchase intention. This is consistent with several prior studies verifying that one's attitude toward a target affects information processing, judgement, and behaviors.

Third, this study considered individualism as a moderating variable. As for the entire circular fashion group, individualism weakened the negative effect of sanitary risk on product attitude and strengthened the positive effect of product attitude on WOM intention and purchase intention. For the secondhand clothing group, individualism strengthened the negative impact of aesthetic risk on product attitude and weakened the negative impact of sanitary risk on product attitude. Regarding upcycled clothing, individualism played a role in moderating the relationship between product attitude and WOM intention as well as in the relationship between product attitude and purchase intention. There was no moderating effect found in the recycled clothing group.

Lastly, most consumers (78.4%) preferred the recycled clothing among the three types of circular fashion. Furthermore, the result of analyzing the differences among the means using multiple comparison indicated that consumers generally perceived higher value and lower risk for recycled clothing compared to secondhand clothing and upcycled clothing. The means of product attitude, WOM intention, and purchase intention of recycled clothing were also significantly higher. Interestingly and perhaps ironically, the perceived environmental value was also found to be the highest for recycled clothing, followed by upcycled clothing and secondhand clothing. This finding was contrary to the actual environmental impact. Given that discarded material is transformed from its original form, the environmental value is high due to low carbon emissions generated during the manufacturing process. However, the actual order of impact is secondhand clothing, upcycled clothing, and recycled clothing. It is interesting to note that despite the lowest environmental value of recycled clothing among the three circular fashion types, consumers' perception was the highest for this category.

## 6. Conclusions

Our study contributes to the literature on circular fashion by enhancing our understanding of the effects of consumers' perceived value and risks of such products. In particular, it is meaningful that this study focused on circular fashion made from textile waste generated in the fashion industry and divided it into three types. It is also meaningful that this empirical study compared differences in consumer perceptions, attitudes, and behaviors by type. This study identified factors affecting circular fashion consumption from a more integrated perspective. Most existing studies related to circular fashion have a limitation in that they have explained purchase based on perceived value using value-attitude-behavior theory. However, this study included not only value dimensions but

also risk dimensions and individualism to examine factors influencing the purchase of circular fashion in a more diverse manner. It is also worth highlighting the significance of perceived emotional value. Emotional value is a unique subjective emotion that consumers feel about the product in the process of purchasing and using it, which, in turn, has a significant impact on purchasing behavior. Given that circular fashion clothing is made of discarded waste, providing eco-friendly information and storytelling such as the product source, functionality, and design plays an important role in forming a good feeling and favorable assessment about circular fashion clothing.

Based on our findings, we recommend consumption revitalization strategies for fashion companies to increase sales of products and provide implications for each type of circular fashion. First, fashion companies should focus on increasing the emotional value of circular fashion for consumers and relieving their anxiety about the sanitary risk. That is, it is a very important challenge for fashion companies to make consumers feel happy and pleased about buying circular fashion products, and to instill a perception that the products are clean even though they are recycled. Second, when selling secondhand clothing, marketers should emphasize that the clothing has been washed and repaired and is clean for resale to reduce consumers' concerns about hygiene. For secondhand clothing, although emotional value, social value, epistemic value, and environmental value were all important factors and had a positive effect on product attitude, the mean value of each variable was the lowest among the three types. More effort is needed to change consumers' perceptions of the value dimensions of secondhand clothing through education or campaign activities. Third, social value and epistemic value of upcycled clothing have a greater positive influence on product attitude compared to the other types. Thus, it is necessary to emphasize the ability to convey a good image to others by wearing upcycled clothing, and to promote more active consumption by emphasizing the uniqueness and novelty of upcycled clothing. Lastly, the results showed that most of the participants preferred recycled clothing among the three types of circular fashion, and the perceived environmental value of recycled clothing had a greater positive effect on product attitude compared to the other types. Despite the increased awareness of circular fashion products, consumers still prefer brand new products to those that have already been used by others. Recycled clothing was preferred by participants in this study over the other categories. They seemed to recognize that recycled clothing made entirely by breaking down discarded material into fibers, making yarn, and making fabric is eco-friendly and are brand new products.

While many studies have focused on altruism and collectivism to determine the moderating role in green consumption, this study sheds new light on the moderating effect of individualism in consumers' sustainable behavior. However, the limitations of this study suggest directions for future research. First, further studies are needed to more elaborately explain why and where individualism plays a significant moderating role. Second, future studies could consider additional factors associated with circular fashion consumption such as past purchase experience, environmental awareness, and perceived skepticism. Third, the study only focused on circular fashion products made of textile waste, hence it is recommended that future research can examine circular fashion products made of other various waste such as PET bottles, fishnets, food waste. Lastly, further studies are required to investigate the role of Hofstede's cultural dimensions and other cultural contexts in affecting consumer attitude and behavior towards circular fashion products since each cultural group is likely to have different levels of circular fashion knowledge and familiarity based on accumulated experiences in respective markets.

**Author Contributions:** Conceptualization, I.K., H.J.J. and Y.L.; methodology, I.K.; software, I.K.; validation, I.K. and Y.L.; formal analysis, I.K.; investigation, H.J.J.; resources, Y.L.; data curation, I.K.; writing—original draft preparation, I.K.; writing—review and editing, H.J.J. and Y.L.; project administration, Y.L.; funding acquisition, H.J.J. All authors have read and agreed to the published version of the manuscript.

**Funding:** This research was funded by the Ministry of Education of the Republic of Korea and the National Research Foundation of Korea (NRF 2019S1A5A2A03054508).

**Institutional Review Board Statement:** Not applicable.

**Acknowledgments:** Our researchers would like to thank anonymous reviewers and editors.

**Conflicts of Interest:** The authors declare no conflict of interest.

## Appendix A

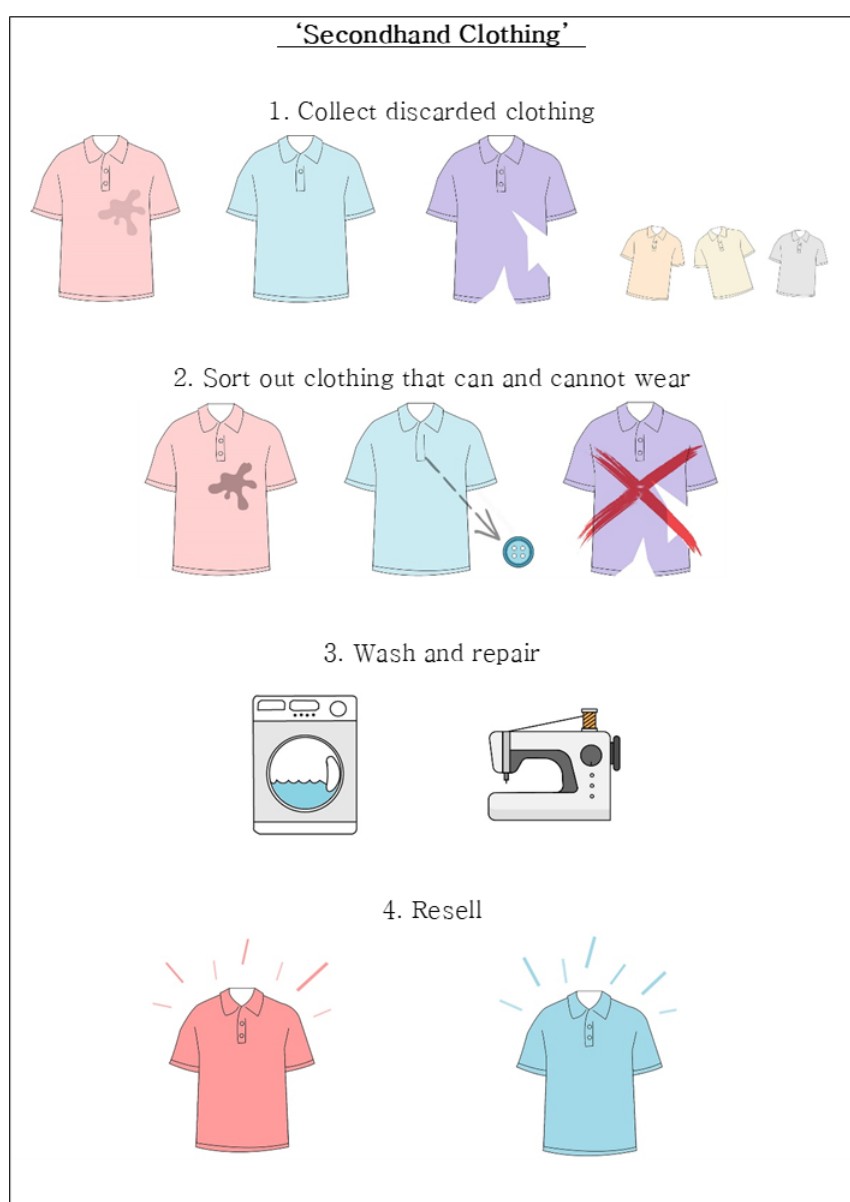

**Figure A1.** Visual Representation of Secondhand Clothing.

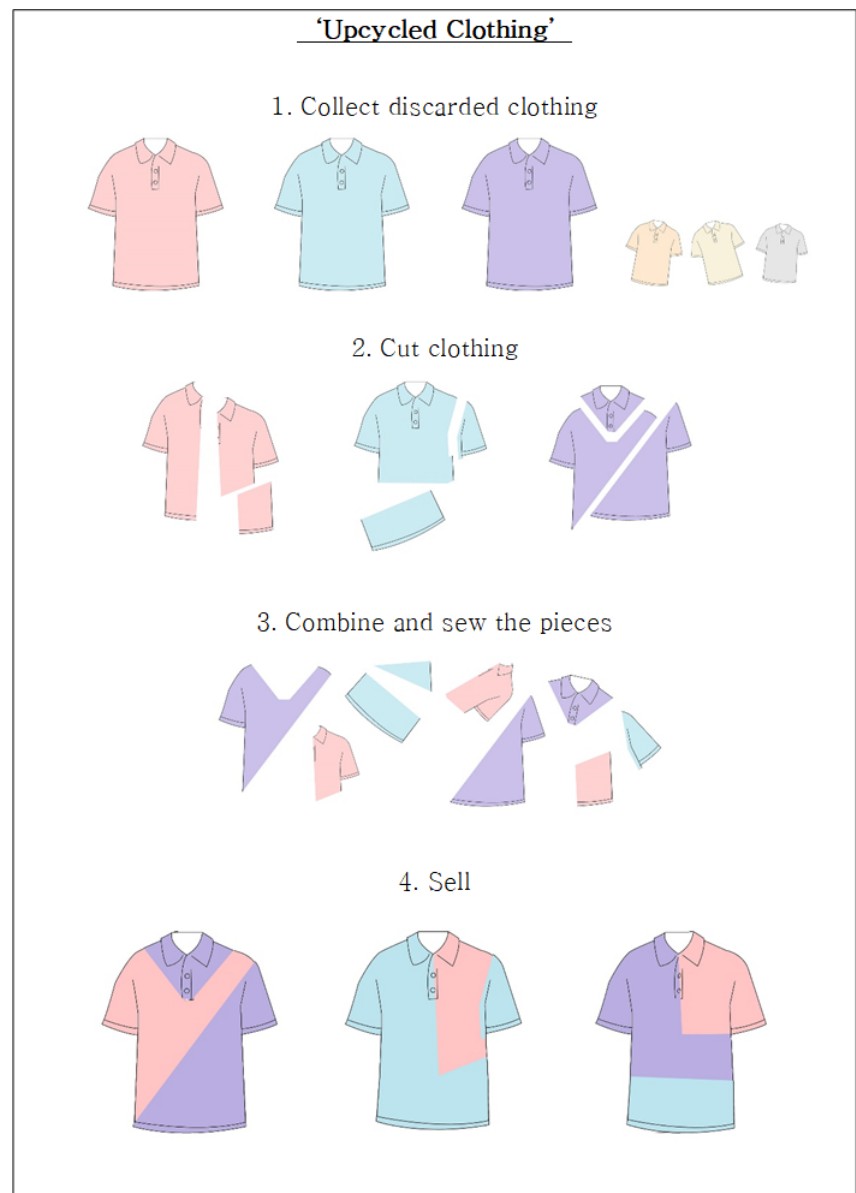

**Figure A2.** Visual Representation of Upcycled Clothing.

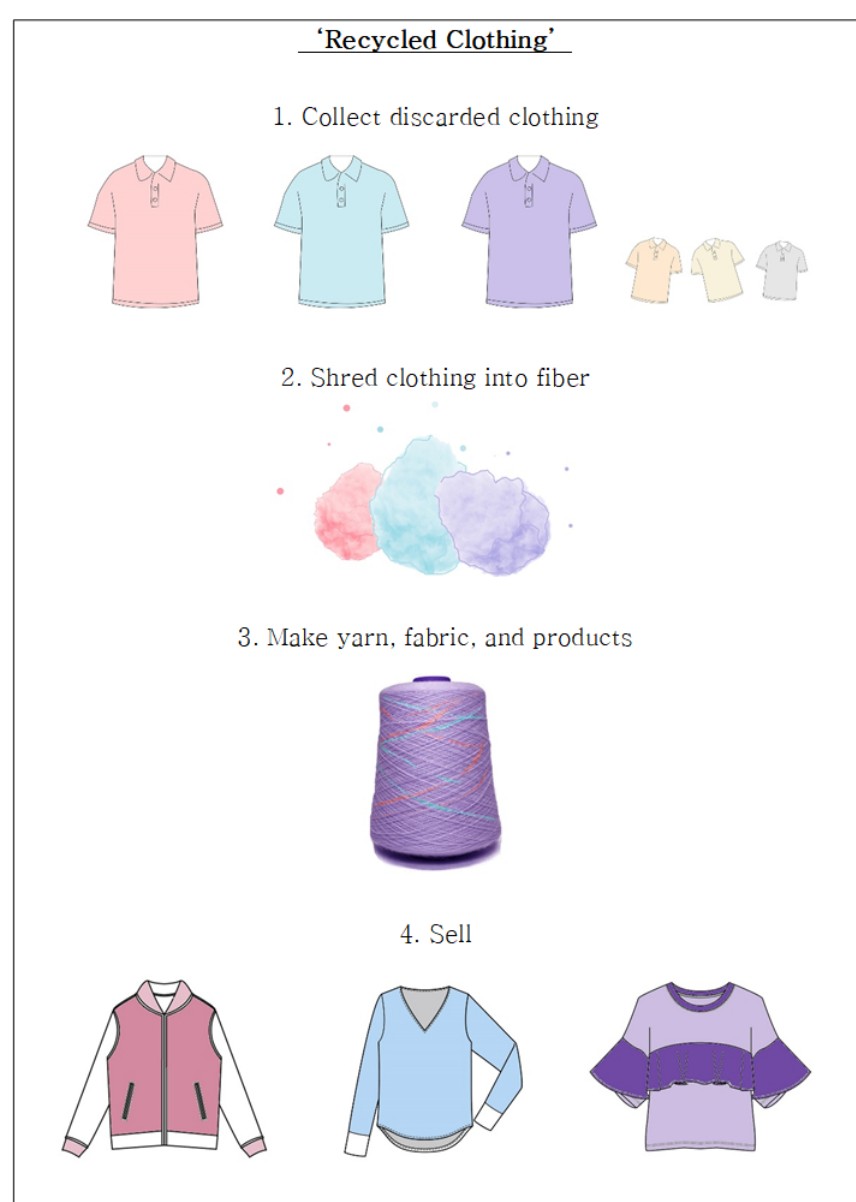

**Figure A3.** Visual Representation of Recycled Clothing.

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
