# Peer review of "Consumers’ Value and Risk Perceptions of Circular Fashion: Comparison between Secondhand, Upcycled, and Recycled Clothing"

_sustainability, doi:10.3390/su13031208_

Round 1
Reviewer 1 Report
The paper is an in-depth and well-founded study on circular fashion by enhancing understanding of the effects of consumers ’perceived value and risks of such products. I recommend publication in the presented form.
Author Response
The authors deeply appreciate the reviewers’ valuable and insightful feedbacks.
Reviewer 2 Report
This manuscript presents an interesting and less researched topic of circular fashion. The purpose of this study was to investigate consumers’ value and risk perceptions, product attitudes, and behavior intentions toward scircular fashion consumption, focusing on reused clothing, upcycled clothing, and recycled clothing.
Overall it is a well written paper presenting an interesting case and could prove valuable in helping retails create marketing strategies and in the more sustainable production cycles.
A few minor comments below:
- At the end of the introduction section, it would be good to provide a short paragraph with the structure of the paper in the sections that follow.
- It would be preferrable to avoid using 'we', 'our' forms and use instead the third tense such as the present research or it was concluded instead of we conclude.
- Finally I would like to see in the discussion or conclusion section a few more details on how this research could be applicable elsewhere or what limitations might occur when applying these results in another country with different cultural characteristics for instance.
Author Response
The authors deeply appreciate the reviewers’ valuable and insightful feedbacks to enhance the depth and width of our manuscript. The authors hope that the revision will be pertinent to your comments and are eager for publication in the Sustainability. The revisions to the manuscript are detailed below:
Comments
1. At the end of the introduction section, it would be good to provide a short paragraph with the structure of the paper in the sections that follow. |
Answer) We added a paragragh to explain the strcture of our paper at the end of the introduction section as follow: “In the following section, the relevant literature review and the hypotheses development are presented. Subsequently, the research methodology is described, followed by the results. Finally, the results are discussed and research contributions, managerial implications, and future research directions are presented.” (p. 2 line 93-96)
2. It would be preferrable to avoid using 'we', 'our' forms and use instead the third tense such as the present research or it was concluded instead of we conclude. |
Answer) We modified them.
We found that -> As a result, (p. 1 line 19)
we expect -> it is expected (p. 2 line 86)
we expect -> it is expected (p. 8 line 389)
we also expect -> it is also expected (p. 9 line 434)
We found no difference -> There was no difference found (p. 12 line 530)
3. Finally I would like to see in the discussion or conclusion section a few more details on how this research could be applicable elsewhere or what limitations might occur when applying these results in another country with different cultural characteristics for instance. |
answer) We added more details based on the reviewer's suggestion as follow: “Lastly, further studies are required to investigate the role of Hofstede’s cultural dimensions and other cultural contexts in affecting consumer attitude and behavior towards circular fashion products since each cultural group is likely to have different levels of circular fashion knowledge and familiarity based on accumulated experiences in respective markets.” (p. 18 line 762-766)